# Dual bronchodilators in Bronchiectasis study (DIBS): protocol for a pragmatic, multicentre, placebo-controlled, three-arm, double-blinded, randomised controlled trial studying bronchodilators in preventing exacerbations of bronchiectasis

Miranda Morton ![ORCID],[1] Nina Wilson ![ORCID],[2] Tara Marie Homer,[3] Laura Simms,[1] Alison Steel,[1] Rebecca Maier,[1] James Wason ![ORCID],[2] Laura Ternent,[3] Alaa Abouhajar,[1] Maria Allen,[4] Richard Joyce,[1] Victoria Hildreth,[1] Rachel Lakey,[1] Svetlana Cherlin,[2] Adam Walker,[4] Graham Devereux,[5] James D Chalmers,[6] Adam T Hill,[7] Charles Haworth,[8] John R Hurst ![ORCID],[9] Anthony De Soyza[2,4]

For numbered affiliations see end of article.

**Correspondence to**
Anthony De Soyza;
anthony.de-soyza@newcastle.ac.uk

## ABSTRACT

**Introduction** Bronchiectasis is a long-term lung condition, with dilated bronchi, chronic inflammation, chronic infection and acute exacerbations. Recurrent exacerbations are associated with poorer clinical outcomes such as increased severity of lung disease, further exacerbations, hospitalisations, reduced quality of life and increased risk of death. Despite an increasing prevalence of bronchiectasis, there is a critical lack of high-quality studies into the disease and no treatments specifically approved for its treatment. This trial aims to establish whether inhaled dual bronchodilators (long acting beta agonist (LABA) and long acting muscarinic antagonist (LAMA)) taken as either a stand-alone therapy or in combination with inhaled corticosteroid (ICS) reduce the number of exacerbations of bronchiectasis requiring treatment with antibiotics during a 12 month treatment period.

**Methods** This is a multicentre, pragmatic, double-blind, randomised controlled trial, incorporating an internal pilot and embedded economic evaluation. 600 adult patients (≥18 years) with CT confirmed bronchiectasis will be recruited and randomised to either inhaled dual therapy (LABA+LAMA), triple therapy (LABA+LAMA+ICS) or matched placebo, in a 2:2:1 ratio (respectively). The primary outcome is the number of protocol defined exacerbations requiring treatment with antibiotics during the 12 month treatment period.

**Ethics and dissemination** Favourable ethical opinion was received from the North East—Newcastle and North Tyneside 2 Research Ethics Committee (reference: 21/NE/0020). Results will be disseminated in peer-reviewed publications, at national and international conferences, in the NIHR *Health Technology Assessments* journal and to participants and the public (using lay language).

**Trial registration number** ISRCTN15988757.

---

**STRENGTHS AND LIMITATIONS OF THIS STUDY**

⇒ Large randomised controlled trial to address unmet clinical need and critical lack of research into bronchiectasis treatments.

⇒ Pragmatic trial design conducted over multiple sites in the UK.

⇒ Challenge of delivering a large multicentre trial in the respiratory disease area in the peri-COVID-19 era.

⇒ Challenge of a static trial design with changing biology of bronchiectasis following shielding behaviours during the COVID-19 pandemic.

---

## INTRODUCTION

Bronchiectasis is a chronic lung condition, characterised by dilated bronchi, leading to symptoms of breathlessness and chronic productive cough, with intermittent infective exacerbations. Bronchiectasis has various potential aetiologies including immune-deficiency syndromes, allergic bronchopulmonary aspergillosis, chronic obstructive pulmonary disease (COPD), ciliary dysfunction and postinfectious, yet studies have found that between a quarter and half of cases are idiopathic.[1 2] Patients often have recurrent, costly hospital admissions, a poorer quality of life[3 4] and clinically significant fatigue.[5 6]

Current estimates suggest a prevalence of 100 000 adult patients with the condition in the UK. Importantly, studies demonstrate that up to 50% of patients with COPD have coexistent bronchiectasis.[7] With approximately 1 000 000 patients with COPD in the UK,[8] there is potential

BMJ

for a significant increase of case-finding of COPD-associated bronchiectasis over the coming years.

Bronchiectasis mortality rates are approximately 50% higher than that of uncomplicated COPD (calculated at 3% per annum) and have been reported to be increasing.[9] Prognosis varies, with a prior study of 91 patients[10] finding that the primary cause of death was usually respiratory, with survival rates of 91% at 4 years and 68.3% at 12.3 years. The same study found factors such as chronic infection with *Pseudomonas aeruginosa* increase mortality.

Multicentre data suggest that the Bronchiectasis Severity Index (BSI), a multicomponent clinical scoring system, is useful in predicting both mortality and morbidity (hospitalisation). This score was validated in 1300 patients across Europe.[11]

Infective exacerbations lead to significant morbidity. Within the UK national audit data patient group, average exacerbation rate is approximately 3 per year (nearly twice the rate of COPD) with an attendant risk of hospitalisation. This is consistent with published American data on the increasing burden of bronchiectasis[12] and in the UK.[13] Previously published UK data also emphasise the burden of bronchiectasis, uncertainties in aetiology and lack of evidence for the treatments that are often used.[14] Hence, improved interventions in bronchiectasis are urgently required.

There is no cure for bronchiectasis. Current modalities of treatment include oral, inhaled or intravenous antibiotics given regularly with additional courses administered for exacerbations. Mucolytics and regular physiotherapy are used to aid sputum clearance and there are additional guidelines for investigation, diagnosis and management of bronchiectasis produced by the British Thoracic Society (BTS).[15]

National audits in the UK suggest up to 80% of patients with bronchiectasis are on inhaled therapy despite limited evidence.[15] Prior studies of inhaled corticosteroids (ICS) in bronchiectasis have shown no clear benefit[15] and are therefore not recommended in prevailing bronchiectasis guidelines. ICS, however, have been shown to have benefits in asthma[16] and in COPD.[17] More recent data emerging from COPD suggests that ICS reduce COPD exacerbation rates but increases pneumonia rates, a side effect that may be very pertinent in bronchiectasis. International guidelines for COPD now suggest a selective application of ICS and that not all patients with COPD benefit from ICS therapy[17]

The role of triple therapy (ICS/LABA/LAMA) and dual bronchodilation (LABA/LAMA) therapy is unclear in bronchiectasis with no robust randomised placebo-controlled data available. Given the likely higher bacterial and inflammatory load in bronchiectasis airways, these therapies need to be specifically studied within this population

## METHODS AND ANALYSIS

Trial methods and analysis are reported as per Standard Protocol Items: Recommendations for Interventional Trials reporting guidelines.[18]

## Trial design

Dual bronchodilators in bronchiectasis study (DIBS) is a pragmatic, multicentre, placebo-controlled, three-arm, double-blinded, randomised controlled trial, incorporating an internal pilot. The trial aim is to recruit a total of 600 adult patients with bronchiectasis from up to 25 secondary care National Health Service (NHS) sites. The trial is Sponsored by The Newcastle upon Tyne Hospitals NHS Foundation Trust (tnu-tr.sponsormanagement@nhs.net).

The internal pilot involves approximately 15 secondary care NHS sites recruiting for a 12 month recruitment period (following activation of the first site to recruitment). Site activation of the NHS sites will be as soon as possible with a staggered approach. The recruitment aim is to recruit 98–125 participants equating to an average recruitment between 1.25 and 1.6 participants recruited per site per month. At the end of the internal pilot, a review will be made by the Trial Management Group (TMG), the Trial Steering Committee (TSC), the Independent Data Monitoring and Ethics Committee (IDMEC) and a consultation with the funder against the following progression criteria to proceed from the internal pilot to the main trial recruitment:

► Average recruitment ≥1.6 participants/month/site activated—continue to main trial and open additional sites (up to 25 sites total)
► Average recruitment ≥1.25 participants/month/site activated—continue to main trial and open additional sites (up to 25 sites total) plus provide an improvement plan after identifying barriers to recruitment through discussion with sites, TMG, TSC and IDMEC as required.
► Average recruitment <1.25 participants/month/site activated—seek further guidance from funder

The TSC and IDMEC are composed of independent experts in the field (clinicians and statisticians) and patient a public representatives. Each committees' roles are defined in their charters.

The main trial phase will follow on from the pilot without a break in recruitment. The main phase of the trial will involve an additional 10 secondary care NHS sites bringing the total number of sites recruiting to the trial to 25 sites. The recruitment target for each site remains 1–2 patients per month throughout the main phase of recruitment which will last for 14 months. The last patient's last visit will be 12 months after the last participant has been recruited.

## Patient eligibility—inclusion and exclusion criteria

Patients are eligible for the trial if all of the following inclusion criteria apply:
1. Adult patients with CT scan confirmed bronchiectasis and bronchiectasis is the predominant primary respiratory disease in the view of the investigator (CT images/CT reports must be available to complete radiological scoring for BSI).

2. History of two or more exacerbations in any 12 month period in the preceding 2 years requiring antibiotics and/or steroids.

3. Evidence of airflow limitation with an forced expiratory volume in 1 s/forced vital capacity ($FEV_1$/FVC) ratio less than 0.7 and/or daily mucus expectoration.

4. Have either (1) less than 20 pack year history of smoking or (2) greater than 20 pack year history of smoking with $FEV_1$ >79% predicted (to exclude COPD).

5. For patients taking ICS, LAMA or LABA treatment prior to recruitment, willing to have these treatments changed or stopped.

6. Stable bronchiectasis with no exacerbations for 4 weeks prior to baseline.

7. Stable dose of oral steroid for 4 weeks prior to baseline (only applicable for patients taking oral steroid as part of standard care).

Patients are excluded from the trial if any of the following exclusion criteria apply:

1. Cystic fibrosis-related bronchiectasis.

2. Where bronchiectasis is not the main disease or there are contraindications to ICS withdrawal.

3. Predominant COPD or asthma.*
   *Patients who have a historical diagnosis of asthma and/or COPD but where the investigator has sufficient evidence to refute these diagnoses can still be included.

4. Indication to remain on ICS (eg, asthma, COPD, allergic bronchopulmonary aspergillosis and inflammatory bowel disease) or known intolerance to any of the trial drugs or their ingredients.

5. Patients with galactose intolerance, total lactase deficiency or glucose-galactose malabsorption.

6. Inability to perform spirometry or quality of life questionnaires.

7. Patients who are:
   a. Pregnant.
   b. Breast feeding.
   c. Of childbearing potential with a positive urine pregnancy test prior to starting trial investigational medicinal product (IMP).
   d. Male or female of childbearing potential unwilling to use contraception throughout the trial (postmenopausal women must be amenorrhoeic for at least 12 months to be considered of non-childbearing potential).

8. Anyone with cognitive impairment who may not be able to consent.

9. Those who do not speak English or cannot comply with trial procedures.

10. Any potential participant who the investigator believes will not be able to complete the trial visits and procedures.

11. A history of allergy or hypersensitivity to any corticosteroid, anticholinergic/muscarinic receptor antagonist, β2-agonist, lactose/milk protein or magnesium stearate or a medical condition such as narrow-angle glaucoma, prostatic hypertrophy or bladder neck obstruction that, in the opinion of the investigator contraindicates trial participation.

12. Use of acute antibiotics or systemic steroids within 4 weeks of baseline (except for antibiotics and/or stable doses of prednisone ≤5 mg used to treat non-respiratory conditions).

13. Malignancy diagnosed within 5 years of the first trial medication administration where the investigator feels the trial may be affected by recurrence or progression of the malignancy (eg, patients with stable breast cancer, current prostate cancer or 'expected curative' cancer surgery) may not be excluded at the investigators discretion.

14. Administration of an investigational agent within 30 days of first dose of trial medication.

### Objectives and outcomes

The primary objective for the trial is to compare the effects of inhaled dual bronchodilators either as a stand-alone dual therapy (LABA+LAMA) or in combination with ICS as a triple therapy (LABA+LAMA+ICS) to placebo on the number of protocol defined bronchiectasis exacerbations experienced per participant over the 12 month trial treatment period. The protocol definition of a bronchiectasis exacerbation is a continued worsening of one or more of symptoms including increased cough, sputum discolouration, excess sputum production, breathlessness and/or fatigue, that require treatment with antibiotic(s).

Additionally, there is a primary economic objective to compare the cost-effectiveness, measured in terms of incremental cost per quality-adjusted life-year (QALY) gained over the 12 month trial treatment period.

Secondary objectives include comparisons between the effects of dual therapy (LABA+LAMA), triple therapy (LABA+LAMA+ICS) and placebo on the number of bronchiectasis-related hospital admissions, the number of emergency hospital admissions, the time to the first exacerbation of bronchiectasis following randomisation, the number of adverse events (AEs) and cessation of treatment, quality of life, breathlessness and lung function, respiratory and cardiac mortality, economic costs and the incremental cost per QALY gained over the patient's life course. Objectives and outcome measures are fully described in table 1.

### Identification, consent and screening of patients

Potential participants aged 18 years or older with CT scan confirmed bronchiectasis, where bronchiectasis is the predominant primary respiratory disease, will be identified and recruited through secondary care NHS sites. Potential participants will be identified by clinicians and members of the research team from patients attending outpatient clinics, spirometry, oxygen services, inpatient admissions and by examination of databases such as local, BronchUK[19] and EMBARC[20] registries. Primary care practices may also be set up as Participant Identification Centres (PICs) to identify potential participants and

**Table 1** Objectives and outcome measures

| Objectives | | Outcome measures |
|---|---|---|
| Primary | Compare effects of inhaled dual bronchodilators either as a stand-alone therapy (LABA/LAMA) or in combination with ICS (ICS/LABA/LAMA; triple therapy) therapy to placebo on the number of protocol defined bronchiectasis exacerbations (per participant) | ► Number of bronchiectasis exacerbations requiring treatment with antibiotics during 12 month treatment period as measured using participant reports and completed weekly exacerbation diary. |
| Primary economic | Compare cost-effectiveness measured in terms of incremental cost-per-QALY gained over the 12 month treatment period. | ► Incremental cost-per QALY at 12 months. Costs based on the cost of the interventions, use of health services via a Healthcare Utilisation Questionnaire administered at baseline, 1, 6 and 12 months postrandomisation and adverse events.<br>► Transport and time for participants to use healthcare appointments will be assessed via the Time and Travel Questionnaire administered at 12 months postrandomisation. |
| Secondary | To compare the effects of dual bronchodilators (LABA/LAMA) and triple (ICS/LABA/LAMA) therapy and placebo on: | |
| | ► Hospital admissions with a primary diagnosis of exacerbation of bronchiectasis | ► Number of hospital admissions for bronchiectasis exacerbations during 12 month treatment period as measured using participant reports and completed weekly exacerbation diary and verified where possible by hospital discharge summary/Hospital Episode Statistics (HES) data.<br>► Hospitalisation due to bronchiectasis exacerbation data collected up to 24 months after visit 1: screening/baseline will be used to extend modelling beyond 12 months as a sensitivity analysis. |
| | ► Time to first exacerbation of bronchiectasis | ► Time to first exacerbation of bronchiectasis as measured using participant reports/completed weekly exacerbation diary. |
| | ► Number of emergency hospital admissions | ► Number of emergency hospital admissions (all cause) as ascertained at 1, 6 and 12 months visits and where needed from primary care records. |
| | ► Adverse events/drug reactions and cessation of treatment | ► Number of adverse events/drug reactions and cessation of treatment as reported by participant to research team and at 1, 6 and 12 months visits. |
| | ► Disease related health status using the St Georges Respiratory questionnaire (SGRQ) and Quality of Life Bronchiectasis (QOL-B) questionnaire | ► SGRQ and QOL-B at baseline, 1, 6 and 12 months visits. |
| | ► Health-related quality of life using the EQ-5D-5L questionnaire | ► EQ-5D-5L at baseline, 1, 6 and 12 months visits. |
| | ► Breathlessness using Baseline and Transition Dyspnoea Indices (BDI and TDI) | ► BDI at baseline and TDI at 1, 6 and 12 months visits. |
| | ► Lung function using spirometry | Postbronchodilator lung function (LABA within 8 hours, short acting beta2 agonist within 2 hours) as measured by spirometry performed to AmericanThoracic Societyy/ European Respiratory Society (ATS/ERS) standards at baseline, 1, 6 and 12 month visits:<br>► Forced expiratory volume in 1 s ($FEV_1$)<br>► Forced vital capacity (FVC) |
| | ► All-cause, respiratory and cardiac mortality | ► As ascertained from medical records or Office for National Statistics (ONS) records of trial participants (collected up to 24 months after visit 1: screening/baseline). |
| | ► Incremental cost per exacerbation avoided | ► Costs based on cost of the interventions (microcosted) and use of health services collected via a Healthcare Utilisation Questionnaire administered at baseline, 1, 6 and 12 months postrandomisation and adverse events collected via case report forms. |
| | ► Costs to the NHS and patients and lifetime cost-effectiveness based on extrapolation modelling | ► Data to populate the model will come from the trial and will be supplemented by HES and ONS data (collected up to 24 months after visit 1: screening/baseline) and by relevant literature and expert opinion and extrapolated over a patient lifetime. |
| | ► Rates of radiologically confirmed pneumonia, compared with participant's normal baseline | ► Number of pneumonia events and total number of participants suffering pneumonia. This will be measured by asking the participants during follow-up visits. |

**Table 1** Continued

| Objectives | | Outcome measures |
|---|---|---|
| Exploratory | ▶ To investigate the relationship between key outcomes (exacerbations and quality of life as measured by SGRQ and QOL-B) with baseline eosinophil (single level recorded at baseline), median eosinophil level (median of last three available recording when not on oral steroids) and baseline BSI | ▶ Eosinophil measurement at baseline and the median of the last three measurements available<br>▶ SGRQ and QOL-B as completed by participants at 1, 6 and 12 months follow-up visits<br>▶ BSI measured at baseline<br>▶ The number of protocol defined bronchiectasis exacerbations requiring treatment with antibiotics during 12 month treatment period as measured using participant reports/participant completed weekly exacerbation diary. |

BSI, Bronchiectasis Severity Index; EQ-5D-5L, 5-level EuroQol5D index; ICS, inhaled corticosteroid; LABA, long acting beta agonist; LAMA, long acting muscarinic antagonist; NHS, National Health Service; QALY, quality-adjusted life-year.

signpost them to an appropriate secondary care recruitment site.

Potential participants will be approached in person or by post with a trial-specific Patient Information Sheet (PIS) (see online supplemental file). Each potential participant will have time to read and consider the PIS before having an opportunity to discuss the trial further with a member of the research team. Written, informed consent using a trial-specific Consent Form (see online supplemental file) will be obtained by a medically qualified member of the research team prior to any trial specific screening activity.

Screening of potential participants will involve:

▶ Collection of demographic data, medical history, medication history and smoking history.
▶ Lung function as measured by spirometry.
▶ Modified Reiff scoring[21] of the latest CT scan performed as part of standard care.
▶ Quality of life measurements including Quality of Life Bronchiectasis (QoL-B) questionnaire,[22] St George's Respiratory Questionnaire (SGRQ)[4] and the 5-level EuroQol5D index (EQ-5D-5L) questionnaire.[23]
▶ BSI score.[11]
▶ Assessments of breathlessness including the Medical Research Council Dyspnoea (MRCD) Score[24] and Baseline Dyspnoea Index (BDI) and Transition Dyspnoea Index (TDI), respectively.[25]
▶ Health Economics questionnaires.
▶ Baseline bloods (full blood count including differentials).

All participants of childbearing must be willing to use an acceptable form of contraception (as listed in the PIS) throughout the trial from the date of consent until 7 days after their last dose of trial medication. Additionally, women of childbearing potential will be required to have a negative urine pregnancy test at baseline/screening to be eligible for the trial.

## Randomisation

Patients who are confirmed as eligible will be randomised to one of three treatment arms dual therapy (LABA+LAMA), triple therapy (LABA+LAMA+ICS) or matched placebo, in a 2:2:1 ratio (respectively). Randomisation is carried out by appropriately delegated members of the research team using random permuted blocks of variable length within strata via the Sealed Envelope system; a central, secure, 24-hour web-based randomisation system with concealed allocation.

Randomisation is stratified by two variables: BSI score (BSI score of 0–8 or 9+) and by baseline ICS drug therapy (ICS user or non-ICS user at baseline).

## Intervention

Treatment allocation is double-blind, each inhaler is identical in appearance to maintain the blind. Participants will have a dose of one inhalation of their trial inhaler per day, it is advised that the dose is taken at the same time of day wherever possible, for the 12 month (365 days) trial period. Dose modifications are not permitted.

The contents of each inhaler are described in table 2.

Each inhaler contains 30 doses of medication. Due to the shelf life of the labelled inhalers, they will be shipped to participants approximately monthly throughout their participation in the trial. Confirmation of receipt of trial medication will be carried out by a member of the research staff telephoning the patient following each trial medication shipment. Trial inhalers are returned by participants at the end of their involvement in the trial and compliance monitored.

## Follow-up visits and participant assessments

Follow-up visits will take place at the local trial site at 1, 6 and 12 months with assessments performed as detailed in table 3.

Participants will be required to complete an Exacerbation Diary weekly throughout their 12 month participation in the trial. In this diary, participants will record whether or not they have experienced any exacerbations during the week and any steps or treatment taken for the exacerbation. This will be reviewed by a member of the research team at each trial visit and collected at the final 12 month follow-up.

Long term follow-up will be carried out 24 months after the screening/baseline visit for participants who reach this time point within the lifetime of the trial. Data collected at this time point are mortality data, the number

**Table 2** Inhaler contents

| Inhaler | Contents of each dose delivered | Name of equivalent commercially available product |
|---|---|---|
| Dual therapy (LAMA/LABA) | ▶ 55 µg umeclidinium<br>▶ 22 µg vilanterol | Anoro Ellipta dry powder inhaler |
| Triple therapy (ICS/LAMA/LABA) | ▶ 92 µg fluticasone furoate<br>▶ 55 µg umeclidinium<br>▶ 22 µg vilanterol | Trelegy Ellipta dry powder inhaler |
| Placebo | ▶ Placebo | Matched placebo dry powder inhaler |

ICS, inhaled corticosteroid; LABA, long acting beta agonist; LAMA, long acting muscarinic antagonist.

of emergency hospital admissions and the number of emergency hospital admissions with exacerbations of bronchiectasis. Participants will not be contacted for this follow-up, instead data will be collected from sources such as patient medical records, BronchUK[19] and/or EMBARC[20] registries or routinely collected Hospital Episode Statistics (HES) and Office of National Statistics (ONS) data.

At the end of their trial participation, participants will stop taking trial medication and continue to have their condition managed through the standard care pathway.

## Pharmacovigilance

All AEs will be recorded in both the participant's medical records and on the electronic Case Report Forms (eCRF) within the trial database. AEs that are judged by an investigator as consistent with the usual clinical pattern for patients with bronchiectasis (such as cough, increased sputum volume and/or consistency, change in sputum colour, wheeze, breathlessness, fatigue and haemoptysis) are not reportable AEs. AEs meeting the seriousness criteria (serious adverse events) will be reported within 24 hours of awareness. Suspected Unexpected Serious Adverse Reactions (SUSAR) will be reported to the MHRA (Medicines and Healthcare products Regulatory Agency) and REC (Research Ethics Committee) within the required regulatory reporting timelines.

Emergency unblinding is available for valid medical or safety reasons where it is necessary for the treating clinician to know which treatment the participant has been receiving. Emergency unblinding should be carried out by the site Principal Investigator or another delegated medically qualified member of the research team by accessing the 24-hour web-based randomisation system.

## Discontinuation and withdrawal

Participants are free to discontinue trial treatment or withdraw from the trial at any time without detriment to their care. Participants choosing to discontinue trial treatment will be invited to complete all trial follow-up visits and assessments. Participants choosing to withdraw from the trial will not take part in any further trial activity. Data collected up to the point of withdrawal will be retained and included in analysis. Participants withdrawn from the trial after they have been randomised will not be replaced.

## Data management

Participant data will be entered by delegated site staff into the trial-specific Clinical Data Management System (CDMS) using Sealed Envelope's Red Pill (Sealed Envelope, London, UK). CDMS users have password limited access restricted to own role and site as appropriate to their delegated duties. Each potential participant will be assigned a unique sequential screening number by site staff, this unique identifier is used to add the participant to the CDMS and becomes their participant ID at the time of randomisation. Overall responsibility for data collection, quality and retention lies with the Chief Investigator who will also hold the final trial data set. Data will be handled, computerised, stored and archived in accordance with the General Data Protection Regulation (2018), and the latest Directive on GCP (2005/28/EC). Newcastle Clinical Trials Unit (NCTU) staff monitor trial conduct and data integrity in accordance with the trial Monitoring Plan. The trial-specific Data Management Plan (DMP) and Data Validation Plan (DVP) include details on how data will be managed and validated throughout the trial.

## Analysis
### Analysis of the primary outcome measure

The primary outcome—number of bronchiectasis exacerbations per participant requiring antibiotics over 12 months—will be compared between randomised groups using negative binomial regression adjusted for stratification factors. This model will be used to compute CIs to test the superiority and non-inferiority hypotheses. The two-sided 95% CI for difference in mean number of exacerbations per year between the combination of LAMA/LABA and ICS/LAMA/LABA compared with placebo will be found: the placebo versus LAMA/LABA hypothesis will be rejected if the upper limit is lower than 0. We will estimate the difference between LAMA/LABA and ICS/LAMA/LABA arms on the relative scale using the incidence rate ratio, and test whether the upper boundary of the two-sided 90% CI is lower than 1.2. We will consider the time at risk to be the time not spent in exacerbation (so that while a patient is in an exacerbation, they are not included as at risk for another).

A secondary analysis will define the at-risk time as the entire length of follow-up for the patient. Estimates will

**Table 3** Schedule of events

| Assessment/activity | Visit 1 Screening/baseline Day 0 | Visit 2 1 month follow-up 1 month (−1/+2 weeks) | Visit 2 Telephone call to participant 3 days (±1 day) following dispensing 2 to 6/7 | Visit 3 6 month follow-up 6 months (−1/+2 weeks) | Visit 3 Telephone call to participant 3 days (±1 day) following dispensing 6/7 to 13 | Visit 4 12 month follow-up 12 months (+2 weeks) | Visit 4 Telephone call to participant 7 days after last dose of IMP (+3 days) | 24 months follow-up |
|---|---|---|---|---|---|---|---|---|
| Written informed consent | X | | | | | | | |
| Demographics | X | | | | | | | |
| Contact details (participant telephone number and address) | X | X | X | X | X | | | |
| Medical history | X | | | | | | | |
| Medication history | X | | | | | | | |
| Smoking history | X | X | | X | | X | | |
| Spirometry including % predicted (FEV$_1$ and FVC) | X | X | | X | | X | | |
| Bronchiectasis Severity Index calculation | X | | | | | X | | |
| Modified Reiff scoring of prior CT scan | X | | | | | | | |
| St George's Respiratory Questionnaire (SGRQ) | X | X | | X | | X | | |
| Quality of Life- Bronchiectasis (QoL–B) | X | X | | X | | X | | |
| Breathlessness: Baseline and Transition Dyspnoea Indices (BDI and TDI) | X | X | | X | | X | | |
| MRC Dyspnoea Score | X | X | | X | | X | | |
| Healthcare Utilisation Questionnaire | X | X | | X | | X | | |
| Time and Travel Questionnaire | | | | | | X | | |
| Quality of life EQ-5D-5L | X | X | | X | | X | | |
| Baseline blood test (FBC with differentials) | X | | | | | | | |
| Pregnancy test (urine, females of childbearing potential) | X | | | | | | | |
| Contraception discussion | X | | | | | | | |
| GP results request/ prior eosinophil levels extracted from medical records/primary care data sets | X | | | | | | | |
| Eligibility confirmation | X | | | | | | | |
| Randomisation | X | | | | | | | |
| Stop prerandomisation inhalers with the exception of short acting beta-agonist (SABA) and commence trial inhalers | X | | | | | | | |
| Trial inhaler delivered to participant on site at face-to-face visit | X | | | | | | | |

**Table 3** Continued

| Assessment/activity | Visit 1 | Visit 2 | | Visit 3 | | Visit 4 | | |
| --- | --- | --- | --- | --- | --- | --- | --- | --- |
| | Screening/baseline | 1 month follow-up | Telephone call to participant | 6 month follow-up | Telephone call to participant | 12 month follow-up | Telephone call to participant | 24 months follow-up |
| | Day 0 | 1 month (−1/+2 weeks) | 3 days (±1 day) following dispensing 2 to 6/7 | 6 months (−1/+2 weeks) | 3 days (±1 day) following dispensing 6/7 to 13 | 12 months (+2 weeks) | 7 days after last dose of IMP (+3 days) | |
| Trial inhaler delivered to participant by post (2–4 days prior to each telephone call) | | | X | | X | | | |
| Confirmation of trial inhaler receipt | | | X | X | X | | | |
| Inhaler technique | X | X | | X | | | | |
| Concomitant medication | X | X | | X | | X | | |
| Issue Patient Exacerbation Diary | X | | | | | | | |
| Participant exacerbation diary review (or reminder to use at telephone calls) | | X | X | X | X | X | | |
| Compliance check and documentation (count of returned trial inhalers and dose counts) | | X | | X | | X | | |
| Number of hospital admissions for bronchiectasis exacerbation | | X | | X | | X | | X |
| Number of emergency hospital admissions | | X | | X | | X | | X |
| Adverse event reporting | | X | X | X | X | X | X | |
| Review of primary care records | | | | | | X | | |
| Mortality data | | | | | | | | X |

EQ-5D-5L, 5-level EuroQol5D index; FEV$_1$, forced expiratory volume in 1 s; FVC, forced vital capacity; GP, general practitioner.

then be adjusted for sites, the stratification factors and other baseline covariates that are known to be strongly related to outcome (eg, age, smoking, bronchiectasis hospitalisations in year prior to trial—these will be prespecified in the Statistical Analysis Plan (SAP)).

We will undertake a sensitivity analysis by excluding those participants who have died. If there is any indication of a differential effect on deaths by treatment, we may consider models that allow the censoring to be informative. For participants who are lost to follow-up by 12 months, their information will be included in the statistical models up to the point that they are lost to follow-up. If loss to follow-up is higher than 10% we will conduct sensitivity analyses to investigate the impact.

We will also explore time to first exacerbation using a Cox regression, and a recurrent events analysis to allow for subsequent exacerbations. In addition, we will use mortality and hospitalisations due to bronchiectasis exacerbation data collected up until 24 months to extend the modelling beyond 12 months as a sensitivity analysis.

For the primary superiority hypothesis, statistical analyses will be according to the intention to treat principle with a per protocol analysis performed as a sensitivity analysis. The per protocol analysis will exclude participants who were not compliant (at less than 75%) with their trial medication or who had a major protocol violation (to be pre-specified in the SAP).

### Analysis of secondary outcome measures

The secondary outcomes: total number of bronchiectasis exacerbations requiring hospital admission, total number of emergency hospital admissions (all causes) will each be analysed as for the primary outcome described above. Disease-related health status (measured using the SGRQ, $FEV_1$ and FVC) will be analysed using a mixed-effects model adjusted for site, stratification factors, patient characteristics and/or baseline clinical variables (to be prespecified in the SAP). Random effects for patient will be included.

### Exploratory and subgroup analyses

Exploratory analyses will investigate the relationship between key outcomes (exacerbations and quality of life as measured by SGRQ and QOL-B) with baseline eosinophils (single level recorded at baseline), median eosinophil level (median of last three available recordings when not on oral steroids) and baseline BSI.

Subgroup analyses of suspected aetiology comparing idiopathic and postinfectious to all other aetiologies for the key outcomes of exacerbations and quality of life (SGRQ and QOL-B) will also be carried out.

All analyses will be governed by this comprehensive SAP which will be agreed by the TSC and reviewed by the IDMEC prior to any analyses being undertaken. Unless prespecified, a 5% two-sided significance level will be used to denote statistical significance throughout.

### Economic evaluation

Both a within-trial and model-based analysis will be undertaken. Both analyses will estimate the incremental cost per QALY gained but over different time horizons; 12 months post-randomisation (within-trial) and over the patients' life course (economic model).

The economic analysis will take the perspective of the NHS and personal and social services. Sensitivity analyses will widen the perspective to incorporate costs incurred by participants. Intervention costs will be estimated based on the manufacturers list price of the medications and inhalers provided to participants. Primary and secondary healthcare resources use will be estimated based on responses to the Healthcare Utilisation Questionnaire (HCUQ) administered at baseline, 1, 6 and 12 months postrandomisation. These data will be combined with unit costs obtained from routine sources to estimate the total healthcare utilisation cost per participant. Additionally, patients with bronchiectasis are likely to experience AEs, treatments and hospitalisations associated with these AEs will be recorded on the eCRF and will be incorporated into the total healthcare utilisation cost per participant in a sensitivity analysis.

Direct (eg, out-of-pocket payments) and indirect (eg, time away from work) costs borne by participants will be collected via the Time and Travel questionnaire administered at 12 months postrandomisation. Similar to the healthcare costs, these study-specific estimates will be combined with routine sources to estimate the total costs incurred for each participant.

Health-related quality of life will be assessed based on responses to the EQ-5D-5L administered at baseline, 1, 6 and 12 months postrandomisation. Utility scores will be derived based on responses to the EQ-5D-5L questionnaire and the national tariff relevant at the time the study reports. QALYs will be estimated using the utility values estimated at baseline, 1, 6 and 12 months using the area under the curve method.[26]

The total cost and QALY per participant will be summarised as the average total cost and average total QALY for each of the randomised arms. All data will be presented as point estimates of incremental costs, QALYs, cost-effectiveness. The incremental cost per QALY gained will be estimated as the difference in costs divided by the difference in QALYs if one of the arms is not dominant (ie, less costly and more effective). The difference in costs and QALYs between the arms will be estimated using seemingly unrelated regression which can control for observed and unobserved characteristics that can affect costs and/or QALYs.[27] Uncertainty in these results, estimated using the bootstrapping technique,[28] will be presented on cost-effectiveness planes and cost-effectiveness acceptability curves (CEACs).

Given that patients with bronchiectasis are expected to take their medications over their life horizon, an economic model will be used to extrapolate the economic results beyond the 12 month treatment period. It is likely

**Table 4** Power calculations

| Scenario | Mean exacerbation rate | | | NI margin (relative to active control) | Power* | |
| | Placebo | ICS/LAMA/LABA (intervention) | LAMA/LABA (active control) | | Non-inferiority (%) | Superiority (%) |
| --- | --- | --- | --- | --- | --- | --- |
| Presented in original grant application | 2.4 | 1.9 | 1.9 | 0.38 (20%) | 90.3 | 89.8 |
| Updated inclusion criteria (same absolute differences) | 1.9 | 1.4 | 1.4 | 0.38 (26.7%) | 96.3 | 95.8 |
| Updated inclusion criteria (same relative differences) | 1.9 | 1.5 | 1.5 | 0.3 (20%) | 83.4 | 83.4 |

*Note this is assuming analytical formulae, with simulations giving consistent but slightly higher powers.
ICS, inhaled corticosteroid; LABA, long acting beta agonist; LAMA, long acting muscarinic antagonist; NI, non-inferiority.

that this model will take the form of a Markov model but the specific form will be decided based on the within-trial results during the model development. Data to design and populate the model will be taken directly from the trial, routine sources (ONS and HES data), literature and clinical opinion. Cost and QALYs incurred after the first 12 months will be discounted at the recommended rate.[29] The model will be developed using the guidance for good practice in conceptualising an economic model.[30] Similar to the within-trial analysis, these results will be presented as point estimates of costs, QALYs and cost-effectiveness. Uncertainty in the economic model will be estimated using Monte Carlo simulations and presented as cost-effectiveness planes and CEACs.[31 32]

### Sample size calculations

The original sample size calculation was based on the number of exacerbations among a similar population in prior national audits.[15 23] The average number of exacerbations was 2.3 per year. Restricting trial entry to those who had ≥3, the original inclusion criteria (assuming a Poisson distribution) gives an average of around 3.8. We therefore assumed, given Hawthorne effect and regression to the mean, that the placebo arm would have a lower mean exacerbation rate than this, and assumed a mean of 2.4 over 1 year. The sample size was chosen so that the trial was well powered to detect a clinically meaningful fall in mean exacerbation rates for bronchiectasis exacerbations to 1.9 year in LAMA/LABA and ICS/LAMA/LABA arms (approximately 20% reduction). This effect size is realistic when compared with the 20%–30% reduction seen in COPD trials with dual bronchodilators/triple therapy and is accepted as clinically meaningful. Although likely studying a different sub population in bronchiectasis, a recent inhaled antibiotic trial in bronchiectasis reduced exacerbations by 39%.[4]

For 90% power (two-sided 5% significance level) to conclude that LAMA/LABA is more effective than placebo with the above parameters, we have calculated

(assuming large-sample approximation of the Poisson distribution) that a sample size of 600 participants is needed, randomised 240:240:120 between LAMA/LABA, ICS/LAMA/LABA and placebo. This allows for a 5% loss to follow-up. This represents a conservative retention rate compared with over 95% observed in the NIHR *HTA* TWICS study with similar pragmatic design and limited patient burden. This calculation assumes a difference between placebo and LABA/LAMA of 0.5 exacerbations per year (21% relative reduction).

If superiority of LAMA/LABA versus placebo is concluded, we will then test non-inferiority of LAMA/LABA against ICS/LAMA/LABA. The sample size will give 90% power (one-sided 5% type I error) with a 0.38 non-inferiority margin (reflecting 20% of the assumed LAMA/LABA rate).

In the early stages of the trial, it was found that exacerbation rates in prebaseline periods were lower than they had been historically, mostly due to the presumed effect of COVID-19 restrictions. The inclusion criteria were therefore updated in protocol V.5.0 (see Amendments). A recalculation of the power of the trial was conducted assuming that there would be lower exacerbation rates in the follow-up period. We assumed that the placebo exacerbation rate would be an average of 1.9 per year. We recalculated the power for two scenarios: (1) assuming that the same absolute difference (0.5) between LAMA/LABA versus placebo and the same non-inferiority margin (0.38) as previously; (2) the same relative difference (21%) between LAMA/LABA vs placebo and relative non-inferiority margin (20%) as before. Table 4 shows the power of the trial to conclude non-inferiority and superiority.

### Amendments

A number of amendments have been approved for the trial. Of note is Amendment 06, a substantial amendment for which the primary purpose was to change eligibility criteria and to extend the pilot phase of the trial.

Initially, the protocol required that patients experience three exacerbations of bronchiectasis within the preceding 12 months to be eligible for the trial. However, this criterion posed a challenge to recruitment given that the biology of bronchiectasis has changed following the shielding behaviour of patients with bronchiectasis during the COVID-19 pandemic. These behaviours have resulted in a reduction in exacerbations being observed in the UK[33] and hence, a reduction in the pool of potentially participants for the trial. In discussion with the oversight committees and funder for the trial, it was agreed to extend the potential participant pool for the trial by changing this eligibility criterion to include patients who have experienced two or more exacerbations within any 12 month period at any time in the preceding 2 years.

As a reflection of the slower recruitment and the challenges faced by sites setting up a respiratory trial during COVID-19, it was agreed to extend the pilot phase of the trial from 6 to 12 months.

## Patient and public involvement (PPI)

Patient and public involvements (PPI) were involved in the trial design and are coapplicants for the research. Independent PPIs have an oversight of the trial and results dissemination as members of the TSC.

## Trial status

The dual bronchodilators in bronchiectasis study (DIBS) trial recruitment opened on 29 July 2021 and closed on 21 October 2022, the last patient's last visit will be in October 2023. This manuscript is based in protocol version 6.0 dated 15 July 2022.

## ETHICS AND DISSEMINATION

The trial received a favourable ethical opinion from the North East—Newcastle and North Tyneside 2 Research Ethics Committee (reference: 21/NE/0020) in March 2021. Trial results will be disseminated in peer-reviewed publications, at national and international academic conferences and in the NIHR *Health Technology Assessments* (HTA) journal. Results will also be disseminated to the public and participants using lay language.

### Author affiliations
[1]Newcastle Clinical Trials Unit, Newcastle University, Newcastle upon Tyne, UK
[2]Population Health Sciences Institute, Newcastle University Faculty of Medical Sciences, Newcastle upon Tyne, UK
[3]Health Economics Group, Newcastle University, Newcastle upon Tyne, UK
[4]The Newcastle upon Tyne Hospitals NHS Foundation Trust, Newcastle upon Tyne, UK
[5]Department of Clinical Sciences, Liverpool School of Tropical Medicine, Liverpool, UK
[6]Molecular and Clinical Medicine, School of Medicine, University of Dundee, Dundee, UK
[7]Centre for Inflammation research, The University of Edinburgh, Edinburgh, UK
[8]Royal Papworth Hospital NHS Foundation Trust, Cambridge, UK
[9]Academic Unit of Respiratory Medicine, UCL Medical School, London, UK

Contributors ADS is the trial Chief Investigator and senior author. AS, RM, CH, JRH and ADS prepared the funding application. MM, NW, TMH, LS, AS, RM, JW, LT, AA, MA, RJ, VH, RL, SC, AW, JC, AH, CH, JRH and ADS have contributed to protocol development and amendments and have revised the manuscript and given approval for the final version. NW, JW and SC have advised on trial design and provide statistical oversight and analysis. TMH and LT have advised on trial design and provide health economic oversight and analysis. MA and AW provide IMP oversight, MM, LS, AS, RM, AA, RJ, VH and RL provide trial and database design and management and trial monitoring.

Funding National Institute for Health Research (NIHR) *Health Technology Assessment* (HTA), funder reference NIHR127460. The views expressed are those of the author(s) and not necessarily those of the NIHR or the Department of Health and Social Care.

Competing interests ADS has received speakers fees or advisory board fees from AstraZeneca, Bayer, GSK, Insmed, Novartis, Gilead and Zambon and has received grants from AstraZeneca, GSK, Novartis and the US COPD Foundation.

Patient and public involvement Patients and/or the public were involved in the design, or conduct, or reporting or dissemination plans of this research. Refer to the Methods section for further details.

Patient consent for publication Not required.

Provenance and peer review Not commissioned; externally peer reviewed.

### ORCID iDs
Miranda Morton http://orcid.org/0000-0002-3362-9302
Nina Wilson http://orcid.org/0000-0001-5908-1720
James Wason http://orcid.org/0000-0002-4691-126X
John R Hurst http://orcid.org/0000-0002-7246-6040

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
