## [Reviewer comments · BMJ Open]

ARTICLE DETAILS

TITLE (PROVISIONAL)	Dual Bronchodilators in Bronchiectasis Study (DIBS): protocol for a pragmatic, multicentre, placebo controlled, 3 arm, double-blinded, randomised controlled trial studying bronchodilators in preventing exacerbations of bronchiectasis
AUTHORS	Morton, Miranda; Wilson, Nina; Homer, Tara; Simms, Laura; Steel, Alison; Maier, Rebecca; Wason, James; Ternent, Laura; Abouhajar, Alaa; Allen, Maria; Joyce, Richard; Hildreth, Victoria; Lakey, Rachel; Cherlin, Svetlana; Walker, Adam; Devereux, Graham; Chalmers, J; Hill, A; Haworth, Charles; Hurst, John; De Soyza, Anthony

VERSION 1 – REVIEW

REVIEWER	Gao, Jinming Peking Union Medical College Hospital
REVIEW RETURNED	26-Feb-2023

GENERAL COMMENTS	Bronchiectasis with airflow limitation is clinically unmet problem. Currently, there is an inconsistency in prescription of bronchodilators, even an argument over long-term management of ICS. To answer these questions is needed. This trial is timely. The trial design and statistics are appropriate. I only have a minor question, that is whether the authors would consider a single bronchodilator (LAMA, LABA) for some patients with mild airway obstruction according to GOLD classification. Excepting from this, I have no conservation to publish this protocol.
--

REVIEWER	van Geffen, Wouter H Medical Centre Leeuwarden
REVIEW RETURNED	05-Mar-2023

GENERAL COMMENTS	This is a multicentre, pragmatic, double blind, randomised controlled trial, incorporating an internal pilot and embedded economic evaluation. The authors aim to include 600 patients and ethical approval has been obtained. Strong points include e.g. the pragmatic design of this trial, the recruiting plan, very professional author team, data magement and high quality endpoints Weak points include 1) the relative high number of patients required per site per month. 2) The exclusion criteria for COPD and asthma patients are a bit vague. COPD is commonly associated with Bronchiectasis and at the one hand they can only be included with impaired FEV1/FVC ratio but should be excluded if the obstructive disease is predominant upon judgement of the recruiting site. This introduces a potential bias especially in case recruiting targets are not met in the process.
---

	3) One might expect different effects depending on the bronchiectasis severity. The authors decided to include also BSI 1-2 patients. Especially when combined with asthma or COPD these might significantly influence treatment effects but reduce clinical relevance of the trial. 4) PFT parameters are difficult to assess when there is bias from asthma or COPD comorbidity in the trial. 5) Patients might already be on ICS or other treatment prior to recruiting. I could not find if study include a wash-out period. A washout period could reduce the bias from prior treatment regarding the endpoint which are exacerbations, as exacerbations may be triggered due to withdrawal of medication rather the effect of the intervention tested.
--	---

VERSION 1 – AUTHOR RESPONSE

Reviewer 1:

- Would the authors consider a single bronchodilator (LAMA, LABA) for some patients with mild airway obstruction according to GOLD classification?

Response: GOLD classification applies to COPD and has no known direct relevance to bronchiectasis. We would be hesitant to treat with only a single long acting bronchodilator as there is increasing evidence to support the additive use of two agents in COPD (2023 GOLD Report - Global Initiative for Chronic Obstructive Lung Disease - GOLD (goldcopd.org) (see table 3.4)). Hence, a negative study in bronchiectasis that included a significant proportion of patients with only one long acting bronchodilator would likely be criticised for minimising the potential of bronchodilation, given the mounting evidence supporting additional benefits of dual agents in COPD (see table 3.4 GOLD Guidance 2023) .

Reviewer 2:

- The relative high number of patients required per site per month

Response: The site target of recruiting 1 to 2 participants per month was thought to be a realistic target in the design of the trial to ensure that the overall recruitment target be met within the recruitment timeframe. Notably many sites across the UK have achieved this level of recruitment in a recent commercial trial in bronchiectasis (unpublished).

Each site undergoes feasibility prior to being selected as a site, within this assessment the site is asked if they envisage any issue with meeting the recruitment target and whether they believe that they can meet it. Only sites who have indicated that they are able to meet the recruitment target are selected to participate in the trial. The trial design incorporates an internal pilot which assesses site recruitment.

- The exclusion criteria for COPD and asthma patients are a bit vague. COPD is commonly associated with Bronchiectasis and at the one hand they can only be included with impaired FEV1/FVC ratio but should be excluded if the obstructive disease is predominant upon judgement of the recruiting site. This introduces a potential bias especially in case recruiting targets are not met in the process.

Response: Both Anoro Ellipta and Trelegy Ellipta are licenced for use in the treatment of COPD with evidence of their effectiveness in this indication. It would be, to our mind, unethical to deny a patient with significant COPD these therapies. Steps were taken in this trial design to exclude those individuals with a significant COPD, or asthma, as their inclusion may have introduced bias on the effect of LAMA/LABA, LAMA/LABA/ICS on the symptoms of bronchiectasis specifically. The inclusion criteria have been designed in line with the pragmatic nature of the trial where a predominant diagnosis will be at the clinical teams discretion. The existing inclusion criteria do allow for patients with normal FEV1/FVC ratio to enter the trial as long as they have daily mucus expectoration. Additionally, patients with a historical diagnosis of COPD, or asthma, are able to enter the trial if there

are sufficient evidence to refute the historical diagnosis. This is a pragmatic effectiveness study designed to replicate day to day clinical practice in the NHS. In our experience of clinical trials of COPD and bronchiectasis that clinicians are able to differentiate between a predominant diagnosis of COPD (with an element of bronchiectasis, usually mild and identified on HRCT) and a predominant diagnosis of bronchiectasis (predominant infective symptoms, widespread cystic changes on HRCT with mild airflow obstruction on spirometry).

We have factored in that there is no test to separate the causality of airflow limitation in bronchiectasis in an ex-smoker vs COPD associated bronchiectasis (see Hurst De Soyza & Elborn ERJ 2015). However, a pragmatic approach allowed site discretion to test these therapies; For an example a patient with severe cystic bronchiectasis, an FEV₁ of 70% predicted and a 15-pack year history is more likely to have primary bronchiectasis as compared to a patients with a 60 pack year history, mild cylindrical bronchiectasis and a FEV₁ of 35% predicted (where COPD associated bronchiectasis is more likely). Collecting these data at baseline allows us to interrogate these difficult questions.

- One might expect different effects depending on the bronchiectasis severity. The authors decided to include also BSI 1-2 patients. Especially when combined with asthma or COPD these might significantly influence treatment effects but reduce clinical relevance of the trial.

Response: We agree that a range of BSI could enter the study based on age, FEV₁ etc. If a trial was not applicable to patients who had bronchiectasis of lower BSI but still exacerbated the study would be criticised for not answering the question does inhaled therapy help this patient subgroup?

Similarly, the study could not be limited to only BSI group 3 as the study if negative would be criticised for limiting the potential for inhaled therapy to help by focusing on only more severe patients.

- PFT parameters are difficult to assess when there is bias from asthma or COPD comorbidity in the trial.

Response: We apologise if this is not clear but significant asthma or COPD are exclusionary factors. PFT parameters in themselves are only partly associated with future exacerbations (as per COPD and the BSI multidomain scoring).

- Patients might already be on ICS or other treatment prior to recruiting. I could not find if study include a wash-out period. A washout period could reduce the bias from prior treatment regarding the endpoint which are exacerbations, as exacerbations may be triggered due to withdrawal of medication rather the effect of the intervention tested.

Response: Our PPI input suggested washout was not a preferred design and indeed pointed out routine practice within the NHS would not take this approach either. Due to this and the pragmatic design of the trial no washout period has been included. Patients may have been taking ICS, LAMA and/or LABA treatment prior to entry to the trial. The inclusion and exclusion criteria have been designed to recruit patients with a stable bronchiectasis and no exacerbations experienced in the 4 weeks prior to trial entry. At screening/baseline details of prior medication are collected, including medication for a respiratory indication. All exacerbations post randomisation will contribute to the primary outcome, however additional analyses to explore the effect having no washout period will be pre-defined in the Statistical Analysis Plan. The trial has an Independent Data Monitoring and Ethics Committee (IDMEC) who are able to review unblinded data. The IDMEC would highlight any trends, such as one arm of patients experiencing exacerbations early in their trial participation and make recommendations to the Trial Steering Committee and the Sponsor to review the design of the trial for either participant safety or data integrity.

Please do not hesitate to get back in touch should any clarification or further information be required. Thank you for your consideration of this manuscript and we look forward to hearing the outcome of the revision.